# The Role of Dysregulated miRNAs in the Pathogenesis, Diagnosis and Treatment of Age-Related Macular Degeneration

**DOI:** 10.3390/ijms23147761

**Published:** 2022-07-14

**Authors:** Karolina Urbańska, Piotr Witold Stępień, Katarzyna Natalia Nowakowska, Martyna Stefaniak, Natalia Osial, Tomasz Chorągiewicz, Mario Damiano Toro, Katarzyna Nowomiejska, Robert Rejdak

**Affiliations:** 1Chair and Department of General and Pediatric Ophthalmology, Medical University of Lublin, 20-079 Lublin, Poland; k.urbanska.98@gmail.com (K.U.); piotr.stepien.dysk@gmail.com (P.W.S.); k.nowakowska98@gmail.com (K.N.N.); martynastefaniakk@gmail.com (M.S.); natalia.osial@gmail.com (N.O.); toro.mario@email.it (M.D.T.); katarzyna.nowomiejska@umlub.pl (K.N.); robert.rejdak@umlub.pl (R.R.); 2Eye Clinic, Public Health Department, University of Naples Federico II, 80131 Naples, Italy

**Keywords:** age-related macular degeneration, miRNA, microRNA, AMD biomarkers, miRNA therapeutics, AMD pathogenesis

## Abstract

Age-related macular degeneration (AMD) is an eye disease causing damage to the macular region of the retina where most of the photoreceptors responsible for central visual acuity are located. MicroRNAs (miRNAs) are small single-stranded non-coding RNA molecules that negatively regulate genes by silent post-transcriptional gene expressions. Previous studies have shown that changes in specific miRNAs are involved in the pathogenesis of eye diseases, including AMD. Altered expressions of miRNAs are related to disturbances of regulating oxidative stress, inflammation, angiogenesis, apoptosis and phagocytosis, which are known factors in the pathogenesis of AMD. Moreover, dysregulation of miRNA is involved in drusen formation. Thus, miRNAs may be used as potential molecular biomarkers for the disease and, furthermore, tailoring therapeutics to particular disturbances in miRNAs may, in the future, offer hope to prevent irreversible vision loss. In this review, we clarify the current state of knowledge about the influence of miRNA on the pathogenesis, diagnosis and treatment of AMD. Our study material consisted of publications, which were found in PubMed, Google Scholar and Embase databases using “Age-related macular degeneration”, “miRNA”, “AMD biomarkers”, “miRNA therapeutics” and “AMD pathogenesis” as keywords. Paper search was limited to articles published from 2011 to date. In the section “Retinal, circulating and vitreous body miRNAs found in human studies”, we limited the search to studies with patients published in 2016–2021.

## 1. Introduction

Age-related macular degeneration (AMD) is an eye disease characterized by damage to the macular region of the retina responsible for central vision acuity [1]. It leads to progressive loss of central vision manifested as blurring or dark spots in the middle part of the visual field and inability to see fine details and colors clearly. AMD is typified by the appearance of drusen in the macula as well as choroidal neovascularization (CNV) or geographic atrophy [2,3]. In 2014 the prevalence of AMD in people aged 45–85 was reported to be 8.69%. Furthermore the incidence of AMD is currently steadily increasing. It is estimated that by 2040 the number of people affected by AMD will increase to 288 million worldwide [4]. AMD remains one of the leading causes of blindness and moderate or severe vision impairment in individuals over the age of 50 years [5]. Although the pathogenesis of the disease is very complex, multifactorial and still not fully understood, it is known that chronic inflammation, oxidative stress and impaired angiogenesis, apoptosis and phagocytosis may contribute to the process [2]. Despite the last two decades having been groundbreaking in treatment of AMD due to the introduction of anti-VEGF injections and photodynamic therapy, to date, there are no effective treatments to prevent progressive irreversible degeneration of light-sensitive cells in the macula, especially in the atrophic form of the disease and the need for a new approach is crucial [3].

MicroRNAs (miRNAs) are small single-stranded non-coding RNA molecules negatively that regulate genes by silent post-transcriptional gene expressions [6,7]. They are present in all cell types and participate in cellular physiology, playing important roles in various biological pathways [8]. Changes in miRNA expression have been investigated in a variety of conditions, including cancers and gastroenterological, neurological or cardiac diseases [9,10,11]. Furthermore, previous studies have shown that specific miRNAs are involved in the pathogenesis of eye diseases, including AMD [12,13,14,15,16].

Aberrant expression of miRNAs in AMD has been confirmed, both in animals and in humans [17]. Altered expressions of miRNAs are related to disturbances of regulating oxidative stress, inflammation, angiogenesis, apoptosis and phagocytosis, which are known factors in the pathogenesis of AMD [8]. Moreover, dysregulation of miRNA is involved in drusen formation [18]. Thus, miRNAs may be used as potential molecular biomarkers for the disease and, furthermore, tailoring therapeutics to particular disturbances in miRNAs may, in the future, offer hope to prevent irreversible vision loss.

The aim of this article is to clarify the current state of knowledge about the influence of miRNA on the pathogenesis and treatment of AMD, although further clinical studies are needed to fully understand their significance in clinical practice. Our study material consisted of publications, which were found in PubMed, Google Scholar and Embase databases. In order to find the proper publications, the search was conducted with the use of a combination of keywords: “Age-related macular degeneration”, “miRNA”, “microRNA”, “AMD biomarkers”, “miRNA therapeutics” and “AMD pathogenesis”. The first step was to find proper publications from the last 10 years. The second step was to carry out an overview of the found publications. Using the phrase “microRNA OR miRNA AND Age-related macular degeneration” and reducing results by years 2011–2021 we found 180 papers. Three were excluded because we searched for articles in English only. For Section 3, Section 4, Section 5 and Section 6, most of the review articles were excluded from the analysis, as we searched for original articles mainly. Most of the articles that were not included in our review did not represent the point of our interests directly. In Section 2 we limited the search to studies with patients and control groups published in 2016–2021. From 142 results, we selected 13 studies, whose target was to detect dysregulation in expression of retinal, circulating or vitreous body miRNAs or to find associations between miRNA expression and polymorphisms in genes associated with AMD.

## 2. Retinal, Circulating and Vitreous Body miRNAs Found in Human Studies (2016–2021)

### 2.1. Retinal miRNA Expression

Retina was used as a material in two studies. Bhattacharjee et al. observed significant up-regulation of miR-34a expression in AMD whole retina and macular region versus age-matched controls [19]. Pogue et al. found that the AMD group had higher expression of miR-7 and the Let-7 cluster, miR-23a and the miR-27a cluster, miR-9, miR-34a, miR-125b, miR-155, miR-146a. These miRNAs are associated with amyloidogenesis, phagocytosis, synaptogenesis, neurotrophic deficits and pro-inflammatory signaling in nervous tissues [20]. miR-34a that was dysregulated in both studies is involved in proinflammatory pathways and has been previously associated with at least 15 neurological, neuro-immune, neuro-inflammatory and/or neurodegenerative pathologies [19].

### 2.2. Vitreous Body miRNA Expression

Only one study used the vitreous body as a material. Menard et al. profiled the miRNA presence in vitreous bodies of patients with wet AMD and the control group. Twenty-six miRNAs had detectable amplification profiles, of which five were characterized as AMD-specific. The research showed an increase in the levels of miR-548a and miR-146a-5p and a decrease for miR-106b, miR-152 and miR-205. These results may suggest that miRNAs have either an impact on AMD pathology or can be produced as a consequence of NV AMD [21].

### 2.3. Circulating miRNA Expression

Most studies were based on circulating miRNAs found in blood plasma or serum. To determine if AMD-specific miRNA can be obtained via less invasive routes than vitreous body samples, Menard et al. profiled levels of miR-146a, miR-106b and miR-152 in plasma. The results were the same as those observed in vitreous humor. Based on the analysis of specificity and sensitivity of the ratio between miR146a/106b, the authors suggest their potential usefulness as the plasma-based biomarkers of NV AMD [21]. Romano at al. compared the levels of miRNA in blood serum of patients with AMD with the levels in retinas and blood samples collected from the AMD animal model (intravitreal injection of Ab amyloid administered to rats’ eyes). They detected six up-regulated miRNAs (miR-9, miR-23a, miR-34a, miR-126, miR-27a, miR-146a) and one down-regulated (miR-155) in case of AMD patients. Three out of these seven were dysregulated both in AMD patients and rats (miR-27a, miR-146a miR-155). The authors suggest that these three miRNAs could represent suitable biomarkers and potential pharmacological targets in AMD [17]. Blasiak et al. analyzed expression of 18 VEGFA gene-regulating miRNAs in the serum of wet AMD patients. Four out of 18 miRNAs had down-regulated expression in AMD patients in comparison to control—miR-34a-5p, miR-126-3p, miR145-5p and miR-205-5p. These miRNAs are related to the regulation of angiogenesis, cytoprotection and protein clearance. Even though the authors showed the limitations of their study, they also emphasized the need to clarify the role of miRNA regulation in AMD pathogenesis [22]. Elbay et al. revealed novel miRNAs with changed expression in wet AMD patients’ plasma in comparison with control. miR-486-5p and miR-626 had higher expression, while miR-885-5p expression was lower. All of these miRNAs are associated with processes dysregulated in the course of AMD—inflammatory responses, endothelial proliferation, neurodegeneration and neovascularization. The authors suggest their usefulness as novel biomarkers for the diagnosis and evaluation of AMD, but they emphasize the need for further studies [23].

### 2.4. Circulating miRNA Expression in Wet and Dry AMD

Some studies were performed using blood samples of both dry and wet AMD patients. Ren et al. found significantly increased expression of miR-27a-3p, miR-29b-3p and miR-195-5p. Up-regulation of these miRNAs was presented both in dry and wet AMD; however, the expression of miR-27a-3p was significantly higher in the wet AMD group. Thus, miR-27a-3p could have a strong possibility of indicating choroidal neovascularization formation. Although changes in blood samples were found, it might not be completely compliant with local changes in the course of AMD. Authors indicate numerous limitations of the study and suggest further investigations to determine the exact role of miRNA in pathogenesis and early detection of AMD [24]. Litwińska et al. selected 13 miRNAs involved in inflammatory pathways and processes that may contribute to AMD and searched for changes in their expressions in blood plasma. The study revealed that wet AMD was an independent factor associated with higher expression of three miRNAs (miR-23a-3p, miR-30b, mir-191-5p) and lower expression of four miRNAs (miR-16-5p, miR-17-3p, miR-150-5p and miR-155-5p). Dry AMD was an independent factor associated with higher expression of five miRNAs (miR-23a-3p, miR-126-3p, miR-126-5p, miR146a, miR-191-5p) and lower expression of three miRNAs (miR-16-5p, miR-17-3p and miR-17-5p). Of six miRNAs whose expressions varied depending on AMD subtype, four (miR-126-3p, miR-126-5p, miR-150-5p and miR-155-5p) were up-regulated in the case of dry AMD and two (miR-30b and miR-191-5p) were up-regulated in wet AMD patients. The authors suggest that these molecules may have an important role in AMD pathogenesis [25]. Ulańczyk et al. selected 14 plasma miRNAs that regulate angiogenesis, inflammatory and cell-survival processes with documented impact on AMD pathogenesis. AMD was determined as an independent factor associated with higher expression of seven miRNAs (miR-23a-3p, miR-126-5p, miR-16-5p, miR-17-3p, miR-17-5p, miR-223-3p and miR-93) and lower expression of two miRNAs (miR-21-3p, miR-155-5p). There were slight differences between wet and dry AMD patients. Authors conclude that miRNAs could be used as potential biomarkers to distinguish between wet and dry AMD [26]. ElShelmani et al. identified 57 miRNAs that were up-regulated in AMD versus control. They found up-regulation of 46 miRNAs in wet AMD, 4 in dry AMD (miR-19a, miR-19b, miR-296-5p, miR-192) and 7 common in both groups, suggesting a potential difference in circulating miRNAs among atrophic and neovascular types. To determine the role of these miRNAs as biomarkers of AMD, they analyzed 14 miRNAs—7 up-regulated in the neovascular group, 5 up-regulated in the atrophic AMD group and 2 miRNAs up-regulated in both groups. They indicated miR-126, miR-410 and miR-19a as potential diagnostic AMD biomarkers. Expression of these miRNAs was significantly increased in serum from AMD patients in comparison to the control group. Additionally, all these miRNAs remained consistent with the pathways that play a role in AMD pathogenesis—angiogenesis, apoptosis and complement dysregulation [27]. In another study ElShelmani et al. detected overexpression of 39 miRNAs in serum of AMD patients that have been previously shown to play important roles in processes related to AMD pathogenesis. Eight miRNAs (hsa-let-7a-5p, hsa-let-7d-5p, hsa-miR-23a-3p, hsa-miR-301a-3p, hsa-miR-361-5p, hsa-miR-27b-3p, hsa-miR-874-3p, hsa-miR-19b-1-5p) had higher expression in the case of dry AMD patients than in wet AMD patients. These differences may increase the promise of potential prognostic value of miRNAs as biomarkers of AMD. The authors suggest that further investigations with larger sample sizes should be performed [28]. El ElShelmani et al. analyzed changes in levels of 377 miRNAs among wet and dry AMD patients and control group. Overexpression of miR-126, miR-19a and miR-410 in both AMD groups was confirmed. Altered expressions of these miRNAs were found to be involved in several pathways, including the VEGF signaling pathway, complement and coagulation cascade, apoptosis and neurodegenerative disorders. The authors conclude that these miRNAs and their target genes had a significant correlation with AMD pathogenesis. As such, they could be potential new targets as predictive biomarkers or therapies for patients with AMD [29].

### 2.5. Relationship between Expression of miRNAs and Polymorphisms in Genes Associated with AMD

miRNAs have an ability to bind specific target mRNAs and induce their translational repression or degradation in response to external stimuli. Polymorphisms within the DNA sequence coding miRNA are able to modify their transcription and binding affinity with target mRNAs. In order to search for variants of miRNA genes contributing to AMD, Strafella et al. selected genes coding miR-146a, miR-31, miR-23a, miR27a, miR-20a and miR-150 for genotyping analysis. Blood samples were obtained in order to extract genomic DNA. Three polymorphisms—rs2910164 (MIR146A), rs11671784 (MIR27A) and rs895819 (MIR27A)—were significantly associated with AMD. Variants of MIR146A may modify interactions with mRNA targets, which may lead to exacerbation of inflammatory and immune responses, whereas variants of MIR27A may activate specific angiogenic pathways. These data suggest that polymorphisms in MIR27A and MIR146A may finally lead to the exacerbation of angiogenic and inflammatory pathways underlying AMD etiopathogenesis. The authors emphasize that further investigations need to be done to validate the real impact of these polymorphisms on the biogenesis, transcription and function of MIR146a and MIR27A, but the availability of miRNAs as the biomarkers can be crucial in better understanding the risk profile for exudative AMD [30]. Ulańczyk et al. [26] analyzed association between plasma miRNAs and polymorphisms in genes linked with AMD (CFH Y402H and ARMS A69S). The authors showed significant correlation for CFH Y402H and lower expression of miR-16-5p. They also suggest its impacts insufficient cell cycle control in patients harboring risk allele, but emphasize that further investigations need to be carried out [26].

### 2.6. Interplay between miRNAs and Physical Examination

It is important to mention that Ulańczyk et al. [26] also estimated the interplay between miRNA expression and changes in physical examination. Central choroidal thickness values were positively correlated with miR-93 expression, whereas thickness of the central retina was positively correlated with miR-16-5p and miR-223-3p expression. MiR-23a-3p expression had a negative correlation with visual acuity. Litwińska et al. found negative correlations between visual acuity and miRNA-191-5p and positive correlations between visual acuity and miRNA-126-3p, -126-5p and -155-5p [25]. This suggests a possible role of miRNAs as prognostic tools.

### 2.7. Summary and Limitations of the Studies

Summing up, the most frequently detected miRNAs in the studies mentioned above were miR-27a, miR-146a, miR-34a, miR-23a, miR-126 and miR-155. These observations may suggest their potential role as biomarkers in AMD. Further investigations are needed to determine the sensitivity and specificity of miRNA-based tests in detection and prognosis courses of AMD. Summary of all miRNAs found in the studies can be found in the Table 1. The limitation of some studies is a small amount of analyzed patients [17,19,20,21,28]. Another limitation is the fact that some studies analyzed miRNAs with influence on AMD pathogenesis proved in the previous studies [17,21,22,25,26,28,30]. This makes it impossible to find novel miRNAs involved in AMD pathogenesis. Analyses of general trends for miRNA abundance and speciation with miRNA PCR arrays were performed in several studies [19,20,23,24,27,29]. These provide us with better insights concerning overall miRNA changes in AMD patients. Only one study provided sensitivity and specificity values of specific miRNAs as potential biomarkers in AMD [24].

## 3. Influence of miRNAs on Angiogenesis in AMD Pathogenesis

Angiogenesis is known to be an important factor in the pathogenesis of AMD. Several miRNAs with significantly changed levels in patients with AMD are thought to be involved in this process. The angiogenic effect is achieved through different mechanisms enhanced by miRNAs’ modulation of physiological and pathological processes. Most miRNAs associated with pathological angiogenesis in AMD influence vascular endothelial growth factor (VEGF) signaling. miR-23 and miR-27 are proven to enhance angiogenic signaling by repression of their target mRNAs encoding antiangiogenic Sprouty2 and Sema6A proteins which negatively regulate MAPK and VEGFR2 signaling in response to angiogenic factors. It was shown that inhibition of these miRNAs impairs angiogenesis in vitro and postnatal retinal vascular development in vivo and suppresses laser-induced choroidal neovascularization in mice [31]. miR-205-5p decreases VEGFA levels through targeting VEGFA 3′UTR-mRNA and also can influence angiogenesis through phosphatidylinositol-3-kinase/protein kinase B—also known as Akt (PI3K)/PKB signaling transduction. Oxidative stress, which is an important factor in AMD, down-regulates miR-205-5p. Consequently, it can increase levels of VEGFA and enhance angiogenesis in AMD patients [32]. miR-106b also regulates choroidal angiogenesis and is decreased in the AMD model due to the unfolded protein response pathway of protein kinase RNA-like ER kinase. It down-regulates the expression of VEGFA and HIF1α. Moreover, it prevents migration of human retinal microvascular endothelial cells and sprouting—cellular processes involved in angiogenesis [33]. miR-93 represses angiogenesis by reducing mRNA and protein expression of VEGF and decreasing human microvascular-endothelial-cell proliferation. This was shown in the experimental laser-induced choroidal neovascularization mice model and human microvascular endothelial cell model [34]. miR-93 as well as miR-302d repress angiogenesis by decreasing VEGFA secretion through interrupting the transforming growth factor beta signaling pathway in ARPE-19 cells. They target TGFBR2 and inhibit transforming growth factor beta-mediated epithelial to mesenchymal transition [35]. Angiogenesis in the AMD seems to be repressed also by miR-126. The study performed in the laser-induced CNV mouse model proved that miR-126 decreases mRNA and protein levels of VEGFA, KDR and SPRED1. It was also shown on human microvascular endothelial cells that up-regulation of miR-126 reduced endothelial cell-tube formation and VEGF-induced migration [36]. miR-302d-3p targets p21Waf1/Cip1, which promotes VEGFA secretion by retinal pigment epithelium (RPE) cells and tube formation of HUVECs, leading to increased choroidal neovascularization [37].

Several miRNAs were shown to influence angiogenesis through pathways independent from VEGF. It is suggested that miR-24 represses angiogenesis by simultaneously regulating multiple components in the actin cytoskeleton pathways. It exerts an inhibitory effect on actin skeleton dynamics, and represses endothelial cell migration, proliferation and angiogenesis by targeting proteins downstream of Rho signaling, including LIMK2, PAK4 and DIAPH1 [38]. It was documented that hsa-mir-361-5p represses angiogenesis through the TGF-β and mTOR pathways. Tube formation assay performed in vitro with antisense treatment of hsa-mir-361-5p revealed significant increase in angiogenesis. It is suggested that hsa-mir-424-5p and hsa-mir-301a-3p have similar effects [39]. miR-150 regulates the transition of macrophages to the AMD-phenotype, leading to macrophage-mediated inflammation and pathologic angiogenesis by down-regulation of stearoyl-CoA desaturase-2 (Scd2) that is independent from VEGF [40]. Summary of miRNAs involved in angiogenesis pathways related with AMD can be found in the Figure 1.

## 4. Influence of miRNAs on Phagocytosis in AMD Pathogenesis

Another important factor in AMD pathogenesis is a disturbance of phagocytic mechanisms, which impairs clearance of Aβ42-peptides from the extracellular space and leads to the formation of drusen [18]. miR-34a was shown to target and repress TREM2 (triggering receptor expressed in myeloid/microglial cells-2) mRNA which is a prominent sensor-receptor of Aβ42 peptide monomers involved in its phagocytosis and homeostatic clearance. Down-regulation of TREM2 leads to the impairment of microglial cells to phagocytose Aβ42 peptide monomers in the central nervous system (CNS) and the formation of drusen. It was also noted that NF-kB and oxidative stress induce repressive influence of miR-34a on TREM2 [19]. The study performed on RPE cells in the two model rats indicated that miR-24 enhances autophagy by targeting and inhibiting CHI3L1, which activates both AKT/mTOR pathways and ERK signaling [41]. Another miRNA involved in the process of phagocytosis is miR-29, which was shown to enhance autophagy by down-regulating p18 and mitigating subsequent lysosomal recruitment of mTORC1. It also repressed p85α, which leads to Akt/mTOR inhibition and promotion of autophagy. Its action is transcription-independent, as was demonstrated with subcellular fractionation [42]. A study measuring photoreceptor outer segments uptake in ARPE19 showed that miR-184 enhance phagocytic activity in human RPE. This is due to down-regulation of a cytoplasmic peripheral membrane protein EZR and increasing LAMP-1 expression. The molecular interaction of EZR with LAMP-1 is required for the formation of phagocytic vacuoles. Inhibition of miR-184 reduces their interaction, and therefore inhibits the formation of phagocytic vacuoles and impairs phagocytic activity, what can possibly lead to development of AMD [43].

## 5. Influence of miRNAs on Apoptosis in AMD Pathogenesis

Apoptosis of retinal cells under oxidative stress conditions is an important factor in AMD pathogenesis. miR-1246 was shown to have a significant role in the inhibition of this process. A peptide thought to protect RPE cells from apoptosis induced by oxidative stress is mini-αA. Inducted expression of the proteins involved in the process of apoptosis, caspase-3 and caspase-14, was reported in the cells incubated with NaIO3. miR-1246 was shown to target these proteins, reduce their levels and thus mediate the anti-apoptotic effect of mini-αA on RPE cells [44]. Another miRNA that protects RPE cells from oxidative injury and apoptosis is miR-626. It targets and down-regulates kelch-like ECH-associated protein 1 (Keap1). Keap1 promotes ubiquitination and proteasomal degradation of nuclear-factor-E2-related factor 2 (Nrf2) through a Cul3-ubiquitin ligase complex. Nrf2 promotes transcription of multiple anti-oxidative genes and detoxifying enzymes. miR-626 levels were shown to be decreased in AMD patients, thus this mechanism of RPE cell protection seems to be an important factor in AMD pathogenesis [45]. hsa-miR-144, which is down-regulated under oxidative stress in RPE, was shown to target and regulate NRF2, which leads to higher RPE cell survival [46].

Fas—an apoptotic factor that activates apoptosis signaling and P53 pathways—is a common target for miR-23a and miR-374a. Pathologic conditions such as oxidative stress lead to an increase in FasL/Fas expression; accordingly, an increased level of Fas was reported in AMD patients and results in RPE apoptosis. At the same time, it was shown that miR-23a and miR-374a are down-regulated in AMD patients’ plasma. miR-23a and miR-374a inhibit Fas up-regulation under oxidative conditions; thus, their overexpression makes RPE cells resistant to oxidative-stress-induced cell death, caspase-3 activity and DNA fragmentation [47,48].

Apoptosis of RPE cells in the pathogenesis of AMD may be caused not only by oxidative stress but also by lipopolysaccharide (LPS). Expression of miR-21-3-p triggered by LPS was shown to take part in the regulation of LPS-induced inflammation and apoptosis. miR-21-3p targets and inhibits RGS4, which leads to aggravation of inflammation and apoptosis by enhancement of LPS-induced production of inflammatory cytokines IL-6 and MCP-1 [49]. Yet another mechanism to induce apoptosis of RPE cells in which miRNA is involved is through exposure to ultraviolet B (UVB) light. miR-340, whose expression is induced by UVB irradiation, was shown to target and inhibit iASPP—an inhibitory factor of p53—and thus to promote apoptosis of RPE cells and suppress cell viability via affecting p53, p21 and caspase-3 protein expression [50].

## 6. Influence of miRNAs on Inflammation and Oxidative Stress in AMD Pathogenesis

Due to high demand of oxygen in the macula region, RPE cells are particularly exposed to oxidative stress, which is one of the determinants of development and progress of AMD. Chronic inflammation, oxidative stress and lipid deposition are also involved in AMD development [2]. miR-146a is under transcriptional control by NF-κB. It has also been stated as up-regulated by ROS and pro-inflammatory cytokines [51]. Up-regulation of miR-146a promotes development of the disease by down-regulating the complement factor H, which represses inflammatory response [52]. The miR-34a molecule has a regulative role in metabolism of lipoproteins. Specifically, it modulates production of apolipoprotein B (apoB)-containing lipoproteins in the liver [53]. Its molecules are over-expressed in RPE and the retina of mice. We can presume that miR-34a down-regulates secretion of apoB in RPE, which dysregulates cellular lipid metabolism. This dysregulation causes deposition of lipids in RPE, which is one of the contributors to AMD development [54,55]. Further research is needed to specify if effects of miR-34a are a possible AMD treatment target. miR-27a was determined as oxidative stress miRNA. It is crucial to determine how miR-27a affects RPE cells by prooxidants. As is known from previous research, overexpression of miR-27a correlates negatively with expression of forkhead box protein O1 (FOXO1). The damage to the retina was aggravated in the FOXO1 gene-knockdown and miR-27a-overexpression groups after exposure to LED but was alleviated in the FOXO1 gene-overexpression or miR-27a-knockdown groups [56]. Figure 2 presents the pathways related with AMD pathogenesis and miRNAs that may influence them. Summary of the most frequently dysregulated miRNAs and their role in AMD pathogenesis can be found in the Table 2.

## 7. miRNAs as Potential Therapeutic Targets

Molecules of miRNA are small, single-stranded, non-coding RNAs, which are evolutionarily conserved. Individual miRNA can regulate multiple genes and pathways, which overall may lead to far-reaching biological outcomes. As many studies have shown that miRNAs may contribute to the AMD pathogenesis, it is important to determine their role as possible treatment targets. miRNA-based therapies involve modulation of pathogenetic pathways by antagonists and mimics (so-called antagomiRs and agomiRs) [57]. Antagomirs contain binding sites complementary to corresponding miRNAs, which inhibits the expression and function of those miRNAs. AgomiRs are synthetic miRNAs that function as the corresponding natural miRNAs [58]. They can be chemically modified double-stranded miRNAs or miRNAs expressed by viral vectors such as retrovirus, lentivirus and adeno-associated virus [57].

Subcutaneous delivery of anti-miR-33 antisense oligonucleotides (ASO) reduced cholesterol deposition in the RPE layer, decreased pathological changes to RPE morphology and reduced immune-cell infiltration in the RPE in aging mice fed with Western-type high fat/cholesterol diet. These findings suggest that miR-33 targeting may decrease cholesterol deposition and ameliorate AMD initiation and progression [59].

Intravitreal injections of miR-142-3p inhibitor in the laser-induced CNV mouse model showed 46% reduction in blood vessel density and a decrease in microglia area of 30%. The influence of miR-142-3p on inhibition of neovascularization and inflammation makes it worth considering as a pothential therapeutic target in AMD [60].

According to Romano et al.’s study, in particular miR-9, miR-23a, miR-27a, miR-34a, miR-146a (which consecutively were up-regulated) and miR-155 (down-regulated) are most likely to be investigated as potential therapeutic targets [17]. MiR-27a, miR-146a and miR-155 have been reported to be associated with inflammatory pathways such as mTOR, TNFα, nuclear factor-kappaB (NF-κB) and HIF signaling.

Even though miRNA-based therapies seem to be promising targets, there are limitations to this method. AMD is a multi-factorial disease, which makes it impossible to point out the single miRNA as the therapeutic target. Targeting an individual miRNA may also lead to a cascade of events resulting in changes in the expression of many genes [58]. It is also important to emphasize that miRNA therapuetics may be unlikely to be present only in the intended target, which may increase the chance of off-target effects [61]. Summing up, miRNA-based therapy will encounter problems that can be classified into three main categories: identifications of the target, delivery and specificity [58]. So far, only several potential miRNA-based therapies have reached phase I and phase II clinical trials [61].

## 8. Potential Role of miRNAs in Clinical Practice

As the papers analyzed in Section 2 are mostly case-control comparisons, we could detect only associations between dysregulated miRNAs and AMD, not causality. That is why we expanded this issue in Section 3, Section 4, Section 5 and Section 6, looking for proofs in the literature that miRNAs may also contribute directly to the AMD pathogenesis. Proofs are based on in vitro or in vivo studies performed on animal models or human cells. Further studies are necessary to estimate their exact influence on AMD pathogenesis, clinical practice and future therapies. Even though optical coherence tomography (OCT) and fluorescein angiography are useful in the diagnosis and monitoring of AMD progression, any novel, non-invasive and highly sensitive test could be groundbreaking. It would be essential to assess if any miRNA dysregulation is detectable before clinical signs of AMD. As some studies reported correlation between changes in physical examination and miRNA expression, it is important to estimate the meaning of miRNAs as tools that could be used in monitoring progression and predicting the course of AMD.

## 9. Conclusions

MiR-27a, miR-146a, miR-23, miR-34, miR-126 and miR-155 were the most frequently dysregulated miRNAs in the studies analyzed above. All of them are associated with processes underlying AMD pathology—inflammation, oxidative stress, angiogenesis and apoptosis. miR-27a, miR-146a and miR-155 are associated with inflammatory pathways such as mTOR, TNF-α, NF-κB and HIF signaling. Additionally, miR-146a is up-regulated by reactive oxygen species, whereas miR-27a affects RPE by prooxidants. Overexpression of miR-27a and miR-23 leads to increased angiogenesis by repression of antiangiogenic mRNAs. Dysregulation of miR-126 induces angiogenesis and inflammatory responses. Overexpression of miR-34 dysregulates lipid metabolism and causes lipids’ deposition in RPE. It also contributes to phagocytosis and drusen formation. Summing up, miRNAs may be used as potential biomarkers and therapeutic targets in AMD, although further studies are needed to fully understand their significance in clinical practice.

## Figures and Tables

**Figure 1 ijms-23-07761-f001:**
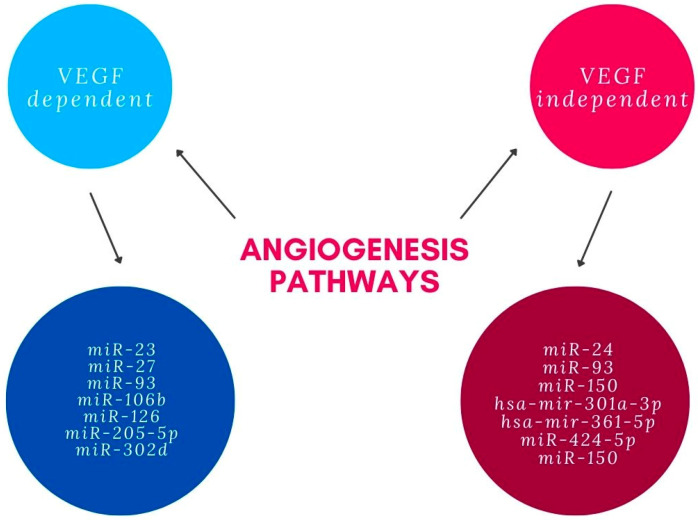
miRNAs involved in angiogenesis pathways.

**Figure 2 ijms-23-07761-f002:**
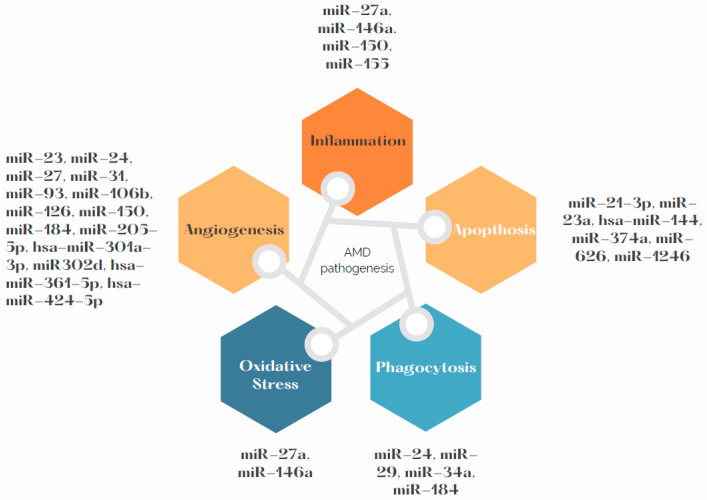
miRNAs involved in AMD pathogenesis.

**Table 1 ijms-23-07761-t001:** Summary of miRNAs found in studies.

Authors	Samples	AMD Group	Control Group	miRNA Expression
Ménard et al. (2016) [21]	Vitreous body	13 patients with wet AMD	13 patients	Increased miR-548a, miR146a-5p Decreased miR-106b, miR-152, miR-205
	Blood plasma			Increased miR-146a Decreased miR-106b, miR-152
Bhattacharjee et al. (2016) [19]	Retina	12 AMD retinas	9 control	Increased miRNA-34a
Ren et al. (2017) [24]	Blood	126 patients with AMD	140 patients	Increased miR-27a-3p, miR-29b-3p and miR-195-5p
Romano et al. (2017) [17]	Blood serum	11 patients with AMD	11 patients	Increased miR-9, miR-23a, miR-27a, miR-34a, miR-126, and miR-146a Decreased miR-155
Pogue et al. (2018) [20]	Retina	17 AMD retinas	10 control	Increased miR-7 and the Let-7 cluster, miR-23a and the miR-27a cluster, miR-9, miR-34a, miR-125b, miR-155, miR-146a
Blasiak et al. (2019) [22]	Blood	76 patients with wet AMD	70 patients	Decreased miR-34a-5p, miR-126-3p, miR-145-5p and miR-205-5p
Litwińska et al. (2019) [25]	Blood plasma	354 patients with AMD (179 with wet AMD, 175 with dry AMD)	121 patients	Wet AMD: increased miR-23a-3p, miR-30b, mir-191-5p, decreased miR-16-5p, miR-17-3p, miR-150-5p and miR-155-5p. Dry AMD: increased miR-23a-3p, miR-126-3p, miR-126-5p, miR-146a, miR-191-5p, decreased miR-16-5p, miR-17-3p and miR-17-5p
Strafella et al. (2019) [30]	Blood	976 patients with wet AMD	1000 patients	Polymorphisms in DNA—MIR146A and MIR27A are significantly associated with AMD
Ulańczyk et al. (2019) [26]	Blood plasma	354 patients (175 patients with dry AMD, 179 patients with wet AMD)	121 patients	Increased miR-23a-3p, miR-126-5p, miR-16-5p, miR-17-3p, miR-17-5p, miR-223-3p and miR-93 Decreased: miR-21-3p. miR-155-5p
Elbay et al. (2019) [23]	Blood—serum	70 patients with wet AMD	50 patients	Increased: miR-486-5p and miR-626 Decreased: miR-885-5p
ElShelmani et al. (2020) [27]	Blood	60 patients (30 patients with dry AMD, 30 patients with wet AMD)	30 patients	46 miRNAs increased in wet AMD group
				4 miRNAs increased in dry AMD
				7 miRNA increased in both groups
				Potential role of miR-126, miR-410, and miR-19a as biomarkers
ElShelmani et al. (2021) [29]	Blood serum	80 patients (40 with wet AMD, 40 with dry AMD	40 patients	Overexpression of miR-126, miR-19a and miR-410
ElShelmani at al. (2021) [28]	Blood serum	26 patients (12 patients with dry AMD, 14 with wet AMD)	10 patients	Increased in dry AMD: hsa-let-7a-5p, hsa-let-7d-5p, hsa-miR-23a-3p, hsa-miR-301a-3p, hsa-miR-361-5p, hsa-miR-27b-3p, hsa-miR-874-3p, hsa-miR-19b-1-5p

**Table 2 ijms-23-07761-t002:** The most frequently dysregulated miRNAs and their role in AMD pathogenesis.

The Most Frequently Dysregulated miRNAs	The Role in AMD Pathogenesis
miR-23	angiogenesis
miR-27a	angiogenesis, inflammation, oxidative stress
miR-34	phagocytosis, drusen formation
miR-126	angiogenesis, inflammation
miR-146a	inflammation, oxidative stress
miR-155	inflammation

## Data Availability

Any additional datasets that are not provided as part of the manuscript are available from the corresponding author on reasonable request.

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
