# Peer review of "The Role of Dysregulated miRNAs in the Pathogenesis, Diagnosis and Treatment of Age-Related Macular Degeneration"

_ijms, 2022, doi:10.3390/ijms23147761_

Round 1

Reviewer 1 Report

In this article, the authors summarized recent research work about the role of miRNA in the pathogenesis, diagnosis, and treatment of AMD. They began by reporting miRNA's expression in retinal, circulating, and vitreous tissues by citing published work, and summarized the miRNAs found in studies in one table. The authors then demonstrated the role of miRNAs on apoptosis, inflammation, and oxidative stress, and as potential therapeutic targets with appropriate cites. Finally, the authors highlighted the most frequently dysregulated miRNAs, which may be used as potential biomarkers and therapeutic targets in AMD. The manuscript is well written. It's very helpful to understand the role of miRNA in AMD.

Author Response

Thank you very much for your opinion. We are glad to get a good opinion on our manuscript from you. 

Reviewer 2 Report

The work of Urbanska and colleagues gives an overview of miRNAs and their role in AMD by looking at studies that compared AMD cases and controls. 

In the introduction AMD is introduced, maybe this can be expanded by telling what it means to loose the central vision. The first sentence is not fully describing AMD to my opinion. 

In line 33 it says "extracellular drusen" suggesting there are also intracellular drusen. What would that be? Please adjust to "drusen" 

in line 42 please explain what  treatment for AMD was groundbreaking in the last decade. 

In line 42, i think it should be contributing instead of contributed. 

Since there is no methods section, please provide at the end of section 1 the exact search terms used so someone can replicate your results. Do not use " a combination of words like" but say exactly what you did. Also mention here that you looked for clinical trials only as you state at the last sentence of the abstract. 

Please also make clear how many papers you have found, how many were not included in this review and the reason why. Where some left out because of quality issues? Were some left out because of only database searches? Maybe make a flow-chart. 

Also try to be as inclusive as you can. I dont see for example the paper of Ertekin 2014 (PMID 25221421)

In section 2 try to be as complete as possible while summarizing the results. For example Bhattacharjee et al, in what did they look for the miRNAs? In blood? In vitreous?  etc. 

TRy to make these sections easy to read, because right now it feels like a large sum up of the papers, while a large part of the information is also written in a table. Make it easier to read by connecting the findings of the studies. Maybe sort the findings, for example by findings in plasma, findings in serum, findings in vitreous etc. 

Make a section at the end of the paper before the conclusion with your own opinion on the findings. For example; why do we need a biomarker for wet AMD for example while we can make a non invasive OCT scan? Besides, most studies are just case-control comparison, not looking at the effects of miRNAs over time on the disease. So only associations not causality. 

How would these miRNAs play a role as a biomarker in clinic? What is your opinion on that? 

In sections 3 to 6 you are loosing me as a reader. What is the message you want to give me? Is it maybe easier to explain with a figure? 

Also miRNAs are mentioned which are not most frequently mentioned by the studies of section number 2. So is the hypothesis that the miRNAs in section 3 to 6 are players in AMD? but not proven?

Author Response

Response to Reviewer 2 comments:

In the introduction AMD is introduced, maybe this can be expanded by telling what it means to loose the central vision. The first sentence is not fully describing AMD to my opinion

We expanded the introduction by adding a description of central vision loss

In line 33 it says "extracellular drusen" suggesting there are also intracellular drusen. What would that be? Please adjust to "drusen" 

We removed ‘extracellular’

in line 42 please explain what  treatment for AMD was groundbreaking in the last decade. 

We specified ‘groundbreaking treatment’ as anty-VEGF injections and photodynamic therapy as they are standard therapies used in treatment over the past decade.

In line 42, i think it should be contributing instead of contributed. 

Corrected

Since there is no methods section, please provide at the end of section 1 the exact search terms used so someone can replicate your results. Do not use " a combination of words like" but say exactly what you did. Also mention here that you looked for clinical trials only as you state at the last sentence of the abstract. 

It was our mistake that we stated that they were clinical trials. They were mostly case-control comparisons. Only one paper (by Ren et al.) is a clinical trial. We changed the end of our abstract.

 Please also make clear how many papers you have found, how many were not included in this review and the reason why. Where some left out because of quality issues? Were some left out because of only database searches? Maybe make a flow-chart. 

We added this information at the end of section 1.

Also try to be as inclusive as you can. I dont see for example the paper of Ertekin 2014 (PMID 25221421)

We didn’t consider this paper, because we analyzed articles from 2016-2021 in this section. However, we added the paper by ElShelmani et al. (2021)

In section 2 try to be as complete as possible while summarizing the results. For example Bhattacharjee et al, in what did they look for the miRNAs? In blood? In vitreous?  etc. TRy to make these sections easy to read, because right now it feels like a large sum up of the papers, while a large part of the information is also written in a table. Make it easier to read by connecting the findings of the studies. Maybe sort the findings, for example by findings in plasma, findings in serum, findings in vitreous etc. 

We have changed this section by sorting the findings by the material used in the studies. That’s why we selected new sections: 2.1-2.5. Section 2.6 is a description of miRNAs’ expression and polymorphisms in genes associated with AMD. Section 2.7 applies to the relationship between changes in physical examination and miRNA expression. We added a 2.8. section concerning summary and limitations of analyzed papers. One new study has been added  

Make a section at the end of the paper before the conclusion with your own opinion on the findings. For example; why do we need a biomarker for wet AMD for example while we can make a non invasive OCT scan? Besides, most studies are just case-control comparison, not looking at the effects of miRNAs over time on the disease. So only associations not causality. How would these miRNAs play a role as a biomarker in clinic? What is your opinion on that? Also miRNAs are mentioned which are not most frequently mentioned by the studies of section number 2. So is the hypothesis that the miRNAs in section 3 to 6 are players in AMD? but not proven?

We added new section (section 8) concerning these issues

In sections 3 to 6 you are loosing me as a reader. What is the message you want to give me? Is it maybe easier to explain with a figure? 

We added a new figure (Figure 1). Figure 2 also concerns the issues from sections 3-6.

Reviewer 3 Report

Dear Authors, 

I wish to submit my review for the Article titled: "The role of Dysregulated miRNA in the Pathogenesis, Diagnosis, and Treatment of Age-related Macular Degeneration.

The paper is well written and provides the readers with a deep analysis of the current knowledge. However, the paper is structured like a systematic review, but definitely, it is not. (Lines 21-26 Abstract)

("Our study material consisted of publications, which were found in PubMed, Google Scholar, and Embase databases. In order to find the proper publications, the search has been conducted with the use of a combination of keywords like: “Age-related macular degeneration”, “miR66 NA”, “AMD biomarkers”, “miRNA therapeutics” and “AMD pathogenesis. The first step was to find proper publications from the last 10 years.")

No chart explains how you conducted the search and which criteria you used to include or exclude the articles. Therefore, it should be considered a narrative review.

Despite the interesting subject, the authors should consider converting it into a systematic review to provide a more rigorous assessment of the existing literature. 

Furthermore, the authors did not include any limitations to this study. (For example, the absence of a rigorous method used to retrieve the papers they included)

Page 2: line 70: The authors listed all the studies. Instead of listing the studies one by one, I think it may be beneficial for the readers to summarize the main concepts and then cite the related articles.

Author Response

Response to Reviewer 3 comments:

The paper is well written and provides the readers with a deep analysis of the current knowledge. However, the paper is structured like a systematic review, but definitely, it is not. (Lines 21-26 Abstract)

("Our study material consisted of publications, which were found in PubMed, Google Scholar, and Embase databases. In order to find the proper publications, the search has been conducted with the use of a combination of keywords like: “Age-related macular degeneration”, “miR66 NA”, “AMD biomarkers”, “miRNA therapeutics” and “AMD pathogenesis. The first step was to find proper publications from the last 10 years.")

No chart explains how you conducted the search and which criteria you used to include or exclude the articles. Therefore, it should be considered a narrative review. Despite the interesting subject, the authors should consider converting it into a systematic review to provide a more rigorous assessment of the existing literature. Furthermore, the authors did not include any limitations to this study. (For example, the absence of a rigorous method used to retrieve the papers they included)

We added this information at the end of section 1.

Page 2: line 70: The authors listed all the studies. Instead of listing the studies one by one, I think it may be beneficial for the readers to summarize the main concepts and then cite the related articles

We have changed this section by sorting the findings by the material used in the studies. That’s why we selected new sections: 2.1-2.5. Section 2.6 is description of miRNA expression and polymorphisms in genes associated with AMD. Section 2.7 applies to the relationship between changes in physical examination and miRNA expression. We added a 2.8. section concerning summary and limitations of analyzed papers. One new study has been added 

Round 2

Reviewer 3 Report

Dear Authors,

Thank you for amending the previously reviewed text.

Although it is not a systematic review, your corrections fulfill the previous requests. I still recommend providing a systematic review according to PRISMA guidelines (Risk Bias assessment, Grade Assessment, ecc.) to strengthen the evidence for future research.